# Exploring the Potential of Foundation Models as Reliable AI Contact Centers

## Author Name

## Abstract

There are several essential requirements for high-quality Contact Centers (CCs). Interalia, correct understanding, courteous interaction, and accurate information provision are crucial. Recently, the advent of foundation models with high generalization performance has brought expectations of potential utilization in CCs applications. Therefore, we explore the feasibility of the foundation models for AI Contact Centers (AICCs). For this purpose, (1) we propose a new dataset for customer service conversations focused on government services in Korea's capital, crafted by experts who work in this service field. (2) We combine audio and text based foundation models to construct the AICC framework. We generate responses about transcribed text from audio with Large Language Models (LLMs) provided prior information to provide factual answers. (3) We evaluate the validity of LLMs answers generated by human evaluators as agent answers. Furthermore, we propose an automatic evaluation method based on LLMs called a generative model-based hierarchical dialog evaluation metric and compare it with the results of human evaluators to further investigate the feasibility of using a foundation model-based evaluation method.

## 1 Introduction

High-quality customer service is an important component of business. In particular, telephone-based customer service (CS) provides the most immediate interaction with customers and resolves customers' issues and queries. However, due to the limited number of human agents, telephone-based CS can easily experience bottlenecks, inevitably leading to delays in service delivery. That is why there is so much interest in applying AI to phone-based customer service for fluent communication and customer-centric problem-solving. AI requires multiple capabilities as a telephone-based CS agent. (1) It must accurately recognize call-based voice data. (2) It should precisely understand the customer's issues and (3) propose appropriate solutions while also being able to use polite and courteous expressions.

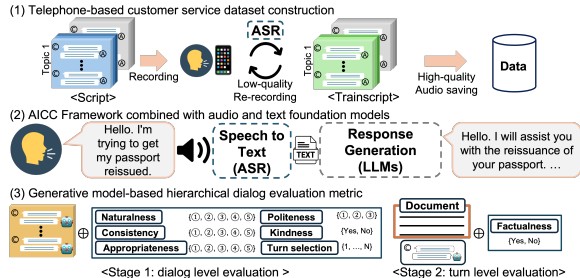

Figure 1: Key contributions of this study. (1) We construct audio data about customer service in the city administration domain through collaboration with domain experts. (2) We exploit the combined foundation models of audio and text as an AI agent. (3) We evaluate the response of the AI agent with our efficient automatic evaluation metric.

To assess the robustness and potential applications of foundation models as AICCs, we constructed novel telephone-based CS data. The prerequisites for data construction are (1) the voices of the speakers are collected taking into account various attributes such as region (e.g., accent, dialect, etc.), gender, and age, and (2) the dialog must contain information that the foundation model hardly learn during pre-training.

The Dasan Call Centre is a telephone service center that handles inquiries and complaints related to the city government. We collected data from the Dasan Call Center that satisfies the above conditions. After collecting the voice data, we improved the data quality by re-collecting samples having a high word error rate (WER) or a character error rate (CER).

We construct the foundation model-based customer service agent in a two-step method. The first step uses Whisper-2 [Radford et al., 2023] to recognize speech data, and the second step uses GPT-4 [Achiam et al., 2023] to generate responses. The middle part of the figure 1 depicts the overall framework. Prompts provided to GPT-4 [Achiam et al., 2023] include the agent's attitude and role, as well as the background knowledge needed for the conversation.

To assess the suitability of the AI agent, we selected six conversational criteria and conducted a human evaluation with them. This method allows for a precise assessment of conversational capabilities.

We propose a generative model-based hierarchical dialog

evaluation metric as an alternative due to the considerable
time and cost of human evaluation. This metric evaluates the
responses of LLMs in dialog across two stages. In the first
stage, we ask LLMs to score each question (e.g., Naturalness,
Politeness, etc.) to evaluate the conversation's comprehensive
quality and select all turns necessary to verify the factualness.
The second stage is to ask LLMs to evaluate the factualness
of the previously selected turns based on relevant documents.
This method efficiently avoids turns that do not require fac-
tual verification allowing for an efficient evaluation process.
We measure the correlation with human judgment and show
our proposed metrics closely correlate with human judgment.
In Figure 1, we depict our key contributions.

## 2 Related Works

### 2.1 Dataset for Auto Speech Recognition

Librispeech [Panayotov *et al.*, 2015] and WHAM [Wich-
ern *et al.*, 2019] are benchmarks for evaluating telephone-
based customer service (ASR) models but do not focus on
task-oriented dialogs or telephone recordings. CALLHOME
[Canavan *et al.*, 1997], on the other hand, consists of tele-
phone conversations. However, it also open-domain dia-
log unsuitable for the AICC dataset. KsponSpeech [Bang
*et al.*, 2020] is one of the large-scale speech corpus of Ko-
rean. While this corpus is an open-domain dialog, ClovaCall
[Ha *et al.*, 2020] is a call-based speech data consisting of a
task-oriented dialog utterance in Korean. Although Clova-
call [Ha *et al.*, 2020] contains short utterance-based record-
ings of restaurant reservation situations, our data consists of
multi-turn scripts and corresponding utterance-based speech
for each scenario, covering one or more administrative tasks
or questions in Korean. In Table 1, we compare the features
with other telephone-based audio datasets. To the best of our
knowledge, Our proposed data is the only telephone-based
city-government service data that considers a combination of
three attributes: accent, gender, and age.

### 2.2 AICC

Much of the previous research on AICC has focused on sup-
porting human agents by performing various tasks in the CC
domain (such as summarizing conversations or determining
intent, etc.) rather than on models that generate answers
based on speech recognition, i.e., direct interaction [Nathan
*et al.*, 2023; Malkiel *et al.*, 2023].

### 2.3 Reference free auto evaluation methods

Traditional reference-based metrics (BLEU [Papineni *et al.*,
2002] and ROUGE [Lin, 2004]) are known to correlate poorly
with human evaluations [Liu *et al.*, 2023; Fu *et al.*, 2023;
Sottana *et al.*, 2023]. There is also research on Langauge
Models to evaluate whether a text summary generated by a
generative model is true based on the given document [Luo
*et al.*, 2023]. We propose a generative model-based evalua-
tion method for response quality and fact-checking, which we
found highly correlated with human judgments.

| Dataset | Lang. | Telephone-based customer service | City government service domain | Utterance-based recording | Attributes balancing | | |
| --- | --- | --- | --- | --- | --- | --- | --- |
| | | | | | Accent | Gender | Age |
| CALLHOME | Eng. | × | × | × | × | × | × |
| FutureBeeAI | Eng. | ✓ | × | × | × | × | × |
| ClovaCall | Kor. | ✓ | × | ✓ | × | × | × |
| Complaint (Call Center) Question-Answer Data | Kor. | ✓ | ✓ | × | × | × | × |
| Ours | Kor. | ✓ | ✓ | ✓ | ✓ | ✓ | ✓ |

Table 1: Comparison of telephone-based audio dataset. The utterance-based recording indicates that audio data exists individually for each speech. Attribute balancing indicates whether audio data is balanced by all attribute combinations.

| | Customer | Agent |
| --- | --- | --- |
| Accent | Standard / Southeastern / Southwestern | Standard |
| Gender | Female / Male | Female / Male |
| Age | Under 50 / Over 50 | Under 50 |

Table 2: Attributes and their categories considered in the dataset.

## 3 Dataset construction and analysis

We provide call-based audio data, Dasan-Call data, for a call-
based customer service task to assess the potential of founda-
tion models to serve as AICC. It consists of scenarios ranging
from a minimum of three to a maximum of five for a total
of 13 topics (e.g., passports, property taxes, etc.) (a total of
56 scenarios). Each scenario script is written based on actual
norms or events, and sensitive information, such as people's
names or phone numbers, has been replaced with arbitrary
values. Additionally, we created a summary of the conversa-
tion for each scenario with experts. We produced additional
versions of each scenario in two different regional (*Yongnam*,
a southeastern province, and *Honam*, a southwestern province
in Korea.) dialects and speech styles besides the standard
language. We collected audio data for all combinations of ac-
cent, gender, and age attributes using scenarios corresponding
to each accent attribute. Table 2 indicates the category of ele-
ments for each attribute we set. For each of the 56 scenarios,
we built audio recording data for 12 attribute groups, result-
ing in a total of 672 voice data. We present the total minutes
of audio data in Table 4. Furthermore, this dataset includes
audio files recorded for each scenario, grouped by attributes,
which enables us to verify whether the ASR model demon-
strates fair performance regardless of the main attributes[1].

## 4 Foundation models for AICCs

To perform a telephone-based customer service task, we se-
quentially use foundation models for both audio and text
modalities. We transcribe the utterer's voice into text by uti-
lizing ASR models (e.g., Whisper-2 [Radford *et al.*, 2023])
and then input this transcribed text into LLMs (e.g., GPT-4
[Achiam *et al.*, 2023]) to generate an appropriate response.
One advantage of using two separate foundation models is
that we can independently select more optimal models for
each task (Speech to Text and response generation).

### 4.1 Auto Speech Recognition

We exploit Whisper-2 [Radford *et al.*, 2023] as an ASR
model, which is based on transformer architecture and trained

---

[1]Our data is published here: https://anonymous.4open.science/r/
AICC_audio_dataset-C2E6/README.md

| | | WER | CER |
|---|---|---|---|
| Accent | Standard | **24.5**\* | **5.8**\* |
| | Southwestern | 49.5\* | 14.4\* |
| | Southeastern | 39.1\* | 11.5\* |
| Gender | Female | **37.6** | **10.3** |
| | Male | 37.8 | 10.8 |
| Age | Under 50 | **36.7** | 10.6 |
| | Over 50 | 38.7 | **10.5** |

Table 3: ASR performance by an element within each attribute. $^*P < 0.05$ (Kruskal-Wallis H-test)

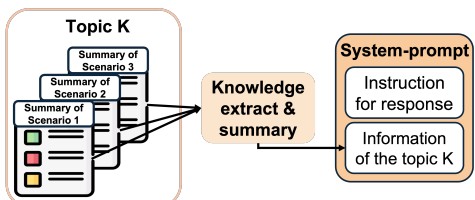

Figure 2: Information extracting from summaries of scenarios in each topic

on a very diverse set of languages and sources. In every scenario, we collect every combination of customer attributes (e.g., accent, gender, and age). Therefore, we assess the fairness of ASR performance on each element of the attributes. In Table 3, we compare how well each attribute is transcribed. Although there are no significant differences in ASR errors by gender and age, accent showed significant performance differences. It is possible that the Korean language learned through Whisper-2 [Radford *et al.*, 2023] includes very little regional dialect or accent, which might explain the significant difference in recognition performance between the standard language and dialects.

## 4.2 Response Generation

We utilized GPT-4 [Achiam *et al.*, 2023] to generate responses to transcribed customer queries. We provide prompts assigning roles (e.g., "Let's assume you're a call center agent.") and guiding the attitude of responses (e.g., "Keep your answers to questions simple, but clear and friendly.") along with the necessary prior knowledge (e.g., documents). As Figure 2 illustrates the system prompt, We gathered scenario summaries for each topic and extracted the information needed for the consultation using LLM. We defined the extracted topic-specific information as prior knowledge and provided it to LLMs as the system prompt. Consistent agent behavior and accurate information delivery are key to enhancing service trust. To achieve reliable responses from LLMs, we not only tried to get precisely crafted system prompts but also adjusted the hyperparameters of GPT-4 [Achiam *et al.*, 2023] to enhance consistency. We set the temperature to 0 and top-P to 1, aiming for possible deterministic answers and expecting high consistency.

## 5 Dialog evaluation

## 5.1 Necessity of response-free evaluations

Prompt engineering optimizes LLM response by guiding reasoning to consistently provide reliable information based on

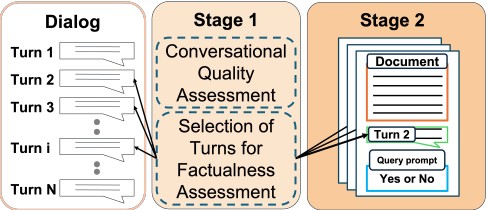

Figure 3: The last question in stage 1 asks participants (human or LLM) to select all the turns that require expertise to verify whether fact. In stage 2, we verify that each turn is true based on expertise.

prior knowledge in various situations. Thus, evaluating LLM responses solely based on references may not be appropriate, as diverse expressions can convey the same intent or information. Therefore, we considered a reference-free evaluation strategy instead. It conducts an evaluation process in two stages: In stage 1, we ask participants to answer the five questions (Naturalness, Consistency, Appropriateness, Politeness, and Kindness) to evaluate whether the agent's responses were appropriately generated throughout the conversation (dialog-level). In stage 2, we evaluate factualness at the turn-level. Table 6 shows the options for each question. Due to time and cost constraints, we sampled 39 scenarios in total, considering all accents per each of the 13 topics, and conducted surveys with two people per sample.

## 5.2 Hierarchical dialog evaluation

We propose a LLMs-based hierarchical dialog evaluation metric, which consists of a 2-stage evaluation. As we depict Figure 3, each stage is divided into assessing the attitude and phrasing of the conversation and assessing the factualness based on prior knowledge. The reason for dividing the stages is that factualness must be verified at the turn level, which requires three elements: prior knowledge, turn, and query prompt. In stage 1, only the entire dialog history and query prompt are necessary. In particular, the prior knowledge required for fact-checking could be large texts (e.g., documents), which can be expensive when using an API for accessing the LLMs. Hence, in stage 2, we only evaluate turns selected for fact-checking in stage 1, in order to perform fact-checking efficiently.

## 6 Experiments & Results

## 6.1 Performance of AICC

As seen in Table 5, the results measured by reference-based metrics are difficult to interpret. Among them, KoBERTScore, which uses KoBERT pre-trained on Korean text data specifically for Korean language processing, quantifies semantic similarity and, therefore, shows a similar tendency to reference-free evaluation. When we evaluate the performance of an AI agent based on human evaluation results, It receives high ratings except for Naturalness. The relatively low evaluation of Naturalness could be due to LLMs' inability to organically connect the information from previous turns when generating a response to the current state's query. Particularly, the Naturalness, Appropriateness, and Factualness performance of the southwestern in accent attributes is

| Utterer | Agent | | Customer | | | | | | | | | | | |
|---|---|---|---|---|---|---|---|---|---|---|---|---|---|---|
| Accent | Standard | | Standard | | | | Southwestern | | | | Southeastern | | | |
| Gender | Female | Male | Female | Male | Female | Male | Female | Male | Female | Male | Female | Male | Female | Male |
| Age | Under 50 | | Under 50 | | Over 50 | | Under 50 | | Over 50 | | Under 50 | | Over 50 | |
| Total min. | 46.5 | 42.8 | 32.8 | 38.9 | 33.4 | 36.7 | 41.6 | 43.6 | 41.1 | 36.7 | 45.5 | 32.3 | 35.4 | 35.6 |

Table 4: Total audio size per attribute group.

| | Reference-based Turn-level Evaluation | | | | | Reference-free dialog-level Human Evaluation | | | | | |
|---|---|---|---|---|---|---|---|---|---|---|---|
| Accent | KoBERTScore (F1) | F1 | BLEU-4 | ROUGE | ROUGE-L | Naturalness | Consistency | Appropriateness | Politeness | Kindness | Factualness |
| Standard | 75.76 ±0.03 | **7.47** ±0.04 | **0.13** ±0.00 | **6.88** ±0.03 | **6.79** ±0.03 | 71.54±18.75 | 90.77±11.41 | 83.08±14.35 | **100.00**±0.00 | **100.00**±0.00 | **86.84** ±0.28 |
| Southwestern | 75.39 ±0.03 | 6.50 ±0.04 | 0.10 ±0.00 | 5.90 ±0.03 | 5.83 ±0.03 | 66.15±20.21 | **93.08**±11.36 | 79.23±13.28 | 98.72±4.44 | 92.31±18.04 | 67.61 ±0.34 |
| Southeastern | **76.09** ±0.02 | 7.17 ±0.04 | 0.07 ±0.00 | 6.48 ±0.03 | 6.40 ±0.03 | **74.62**±24.06 | 87.69±11.87 | **84.62**±17.37 | **100.00**±0.00 | **100.00**±0.00 | 82.67 ±0.19 |

Table 5: Results of AI agent (ASR+Response generation) performing the customer service task with our dataset. We quantify reference-free criteria with the human survey results. All scores are converted to a percentage.

| Criteria | Naturalness | Consistency | Turn selection |
|---|---|---|---|
| {answer-choice} | {1,2,3,4,5} | {1,2,3,4,5} | turn-{1, . . . , N} |
| Appropriateness | Politeness | Kindness | Factualness |
| {1,2,3,4,5} | {1,2,3} | {Yes, No} | {Yes, No} |

Table 6: The Naturalness asks how realistic and smooth the conversation is. Consistency asks whether the agent's responses remain stable regarding opinions and information. Appropriateness asks whether the agent's responses are relevant and logical. Politeness and Kindness ask whether the use of formal language and the tone of responses, respectively. The turn that needs to be verified before the turn-level fact check is selected.

| LLMs | Lang. | Pearson | Spearman | Kendall |
|---|---|---|---|---|
| Llama-3 | Eng. | 77.34 | 69.18 | 61.34 |
| | Kor. | 83.61 | 75.57 | 67.25 |
| GPT-4 | Eng. | 88.74 | 83.91 | 74.10 |
| | Kor. | 89.32 | 84.48 | **75.62** |
| GPT-4-Ensemble | | **90.10** | **85.40** | 74.76 |

Table 7: Correlation between human and LLM judgment results in stage 1. GPT-4-Ensemble represents the average of GPT-4 results queried in English and Korean.

| | | AUROC | |
|---|---|---|---|
| LLMs | Lang. | Human Union | Human Intersection |
| Llama-3 | Eng. | 0.659 | 0.668 |
| | Kor. | 0.726 | **0.735** |
| GPT-4 | Eng. | 0.709 | 0.721 |
| | Kor. | 0.711 | 0.718 |
| Ensemble-inter | | 0.709 | 0.709 |
| Ensemble-union | | **0.773** | 0.729 |

Table 8: Accuracy of LLMs based on human-annotated turns for factuality checking within dialogs. Human Union (Ensemble-union) denotes considering all turns that are selected by at least one participant (LLM). Human intersection (Ensemble-inter), in contrast, considers the turns chosen by all participants (LLMs).

| | | Human Union Label | | Human Inter. Label | |
|---|---|---|---|---|---|
| LLMs | Lang. | ACC. | AUROC | ACC. | AUROC |
| Llama-3 | Eng. | 51.80 | 58.36 | 53.15 | 54.43 |
| | Kor. | 58.11 | 56.72 | 54.05 | 53.02 |
| GPT-4 | Eng. | 59.01 | 63.71 | 52.25 | 52.33 |
| | Kor. | 63.06 | **69.48** | 57.21 | **57.37** |
| Ensemble-inter | | 43.69 | 62.08 | 48.65 | 53.13 |
| Ensemble-union | | **72.07** | 60.71 | **59.91** | 55.46 |

Table 9: Accuracy for the factualness of LLMs based on a human judge in stage 2 for selected turns. Human Union Labels denotes 1 for all turns that are determined to be fact by at least one participant and 0 for others. Human intersection (Ensemble-inter) Labels, in contrast, consider 1 for the selected turns when all participants annotated them as fact and 0 otherwise.

relatively low, which could be due to the influence of ASR results on the response generation of LLMs. We also evaluate the factualness of an AI agent considering the result of the human evaluators' assessment to be the true label.

## 6.2 Correlation with human evaluation

We also conduct hierarchical dialog evaluation with LLMs. We utilize GPT-4 [Achiam *et al.*, 2023] and Llama-3 [Touvron *et al.*, 2023], representative state-of-the-art open-source and closed-source LLMs, respectively. We prepare input prompts in two languages: English, the major language of the pre-training data, and Korean, the language used in the dialog. In Table 7, we compare the result of stage 1 evaluation of LLMs with human judgments. Both LLMs showed a higher correlation when the input prompt was in the same language as the dialog. We observed that the ensemble of results obtained from the two different language versions of GPT-4 [Achiam *et al.*, 2023] better correlated with human answers. This implies that we could consider advanced ensemble methods as a more reliable automatic evaluation method. Table 8 shows how accurately it chooses the turn for fact-checking based on human judgment. Table 9 shows how accurate the

factualness assessment of LLMs based on human judgment is. In Turn selection and factualness evaluation methods, GPT-4 [Achiam *et al.*, 2023] over Llama-3 [Touvron *et al.*, 2023], and it performed better when the input prompt is Korean rather than English.

## 7 Conclusion

We have developed a telephone-based customer service dataset specialized in city government to explore the potential application of foundation models as AI agents. We found that accent features significantly impact ASR performance, which, in turn, can affect conversation quality. We verified that foundation models can perform well as agents with brief instruction and prior knowledge. Moreover, we propose a hierarchical dialog evaluation method based on LLMs that is efficient and similar to human judgment.

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
