# OpenReview forum: "Exploring the Potential of Foundation Models as Reliable AI Contact Centers"
_ijcai.org/IJCAI/2024/Workshop/TIDMwFM — IJCAI TIDMwFM 2024 Oral_

### Official Review · Reviewer_GabR · 2024-06-19

**Rating:** 7
**Confidence:** 4

**Review:**

This paper investigates the integration of foundation models, specifically Whisper-2 and GPT-4, to develop AI Contact Centers for city government services in Korea. By creating a novel, balanced dataset of customer service conversations and employing a two-step model integration process, the authors provide a comprehensive framework for assessing the feasibility of these AI models in real-world applications.

The study's methods are robust, involving both human and automated evaluations. The results indicate that while Whisper-2's ASR performance varies significantly with accents, GPT-4 generates accurate and polite responses, though it struggles with naturalness due to context maintenance challenges. The proposed automatic evaluation metric, which shows a high correlation with human judgments, is a notable contribution that could streamline the evaluation process of AI systems.

This research is highly relevant to the theme of "Trustworthy Interactive Decision-Making with Foundation Models Workshop," as it demonstrates the potential of foundation models in delivering reliable and courteous customer service, essential for building trust in AI systems. The paper's novel dataset and evaluation metric further enhance its contribution to the field.

---

### Official Review · Reviewer_DbdN · 2024-06-21

**Rating:** 8
**Confidence:** 3

**Review:**

Summary: This paper investigates the feasibility of using foundation models for AI Contact Centers, focusing on government services in Korea. The authors propose a new dataset for customer service conversations, combine audio and text-based foundation models to construct an AICC framework, and evaluate the system using both human evaluators and a novel automatic evaluation method based on LLMs.

Strengths: The authors create a new, specialized dataset for customer service in the government domain, considering various attributes like accent, gender, and age. The study combines both speech recognition and text generation models, providing a complete pipeline for an AICC system. Moreover, the proposed generative model-based hierarchical dialog evaluation metric offers an efficient alternative to human evaluation.

Feedback to authors: Consider providing a comparison of your AICC system with existing solutions or baseline models to better contextualize its performance. A more detailed analysis of the LLM's performance in generating responses, including its strengths and limitations could strengthen the paper.